# Genetic Variants in Antineutrophil Cytoplasmic Antibody-Associated Vasculitis: A Bayesian Approach and Systematic Review

**DOI:** 10.3390/jcm8020266

**Published:** 2019-02-21

**Authors:** Kwang Seob Lee, Andreas Kronbichler, Daniel Fernando Pereira Vasconcelos, Felipe Rodolfo Pereira da Silva, Younhee Ko, Yeon Su Oh, Michael Eisenhut, Peter A. Merkel, David Jayne, Christopher I. Amos, Katherine A. Siminovitch, Chinar Rahmattulla, Keum Hwa Lee, Jae Il Shin

**Affiliations:** 1Severance Hospital, Yonsei University College of Medicine, Seoul 03722, Korea; kwangseob@yuhs.ac; 2Department of Internal Medicine IV (Nephrology and Hypertension), Medical University Innsbruck, 6020 Innsbruck, Austria; andreas.kronbichler@i-med.ac.at; 3Laboratory of Histological Analysis and Preparation (LAPHIS), Federal University of Piaui, Parnaiba 64202-020, Brazil; vasconcelos@ufpi.edu.br (D.F.P.V.); feliperodolfo.15@hotmail.com (F.R.P.d.S.); 4Division of Biomedical Engineering, Hankuk University of Foreign Studies, Gyeonggi-do 17035, Korea; younko@hufs.ac.kr; 5Yonsei University College of Medicine, Seoul 03722, Korea; nasaoh@naver.com; 6Department of Pediatrics, Luton & Dunstable University Hospital NHS Foundation Trust, Luton LU4 0DZ, UK; michael_eisenhut@yahoo.com; 7Division of Rheumatology, Department of Medicine, University of Pennsylvania, Philadelphia, PA 19146, USA; Peter.Merkel@uphs.upenn.edu; 8Department of Biostatistics, Epidemiology, and Informatics, University of Pennsylvania, Philadelphia, PA 19146, USA; 9Vasculitis and Lupus Clinic, Addenbrooke’s Hospital, Cambridge CB2 0QQ, UK; dj106@cam.ac.uk; 10Department of Medicine, University of Cambridge, Cambridge CB2 0QQ, UK; 11Institute for Clinical and Translational Research, Baylor College of Medicine, Houston, TX 77030, USA; Christopher.I.Amos@dartmouth.edu; 12Mount Sinai Hospital, Lunenfeld-Tanenbaum Research Institute, Toronto General Research Institute and University of Toronto, Toronto, ON M5G 1X5, Canada; Katherine.Siminovitch@sinaihealthsystem.ca; 13Department of Pathology, Leiden University Medical Center, 2300 RC Leiden, The Netherlands; c.rahmattulla@lumc.nl; 14Department of Pediatrics, Yonsei University College of Medicine, Seoul 03722, Korea; AZSAGM@yuhs.ac; 15Department of Pediatric Nephrology, Severance Children’s Hospital, Seoul 03722, Korea; 16Institute of Kidney Disease Research, Yonsei University College of Medicine, Seoul 03722, Korea

**Keywords:** antineutrophil cytoplasmic antibody (ANCA), proteinase 3, myeloperoxidase, single nucleotide polymorphism, vasculitis, meta-analysis, genome-wide association study

## Abstract

A number of genome-wide association studies (GWASs) and meta-analyses of genetic variants have been performed in antineutrophil cytoplasmic antibody (ANCA)-associated vasculitis. We reinterpreted previous studies using false-positive report probability (FPRP) and Bayesian false discovery probability (BFDP). This study searched publications in PubMed and Excerpta Medica Database (EMBASE) up to February 2018. Identification of noteworthy associations were analyzed using FPRP and BFDP, and data (i.e., odds ratio (OR), 95% confidence interval (CI), *p*-value) related to significant associations were separately extracted. Using filtered gene variants, gene ontology (GO) enrichment analysis and protein–protein interaction (PPI) networks were performed. Overall, 241 articles were identified, and 7 were selected for analysis. Single nucleotide polymorphisms (SNPs) discovered by GWASs were shown to be noteworthy, whereas only 27% of significant results from meta-analyses of observational studies were noteworthy. Eighty-five percent of SNPs with borderline *p*-values (5.0 × 10^−8^ < *p* < 0.05) in GWASs were found to be noteworthy. No overlapping SNPs were found between PR3-ANCA and MPO-ANCA vasculitis. GO analysis revealed immune-related GO terms, including “antigen processing and presentation of peptide or polysaccharide antigen via major histocompatibility complex (MHC) class II”, “interferon-gamma-mediated (IFN-γ) signaling pathway”. By using FPRP and BFDP, network analysis of noteworthy genetic variants discovered genetic risk factors associated with the IFN-γ pathway as novel mechanisms potentially implicated in the complex pathogenesis of ANCA-associated vasculitis.

## 1. Introduction

Antineutrophil cytoplasmic antibody (ANCA)-associated vasculitis (AAV) is a group of autoimmune diseases characterized by the inflammation of small vessels. Clinical overlap among the different subtypes of AAV, granulomatosis with polyangiitis (GPA), microscopic polyangiitis (MPA) and eosinophilic granulomatosis with polyangiitis (EGPA) can result in diagnostic difficulties [1]. The clinical manifestations of the disease types can vary widely; most patients have signs of a lung, kidney, and/or ear, nose, and throat (ENT) involvement. ENT involvement is especially common in patients with GPA and EGPA [1]. A positive test for ANCA aids in the initial diagnosis of AAV. However, overlaps between disease phenotype and ANCA serotype limit the ability of the ANCA serotype to discriminate between the different disease phenotypes [2]. In general, a perinuclear ANCA (p-ANCA)—target antigen MPO—is found in 60–80% of MPA patients. On the other hand, cytoplasmic ANCA (c-ANCA)—target antigen proteinase 3 (PR3)—is frequently found in patients with severe GPA (approximately 90% of patients) and to a lesser degree in those with non-severe disease (approximately 50% of patients) [1].

Even though the exact pathogenesis of AAV is unclear, it is generally believed that AAV has a genetic background. Several genome-wide association studies (GWASs) and meta-analyses of observational studies employing single nucleotide polymorphisms (SNPs) revealed a number of genetic variants to be associated with AAV [3,4,5,6]. Up to date, there have been three GWASs and meta-analyses with replication cohorts: two by the US Vasculitis Clinical Research Consortium (VCRC) [4,6] and one by the European Vasculitis Genetic Consortium (EVGC) [3]. Moreover, Rahmattulla and co-workers performed a meta-analysis with inclusion of EVGC GWAS data and other meta-analyses of observational studies [5].

Because the prior probabilities of genetic associations are low, the number of false-positive associations that are generated by chance alone is high. Not accounting for these low probabilities in the statistical analysis leads to an increased likelihood of finding false-positive associations. Since concerns have been raised about true positives of the discovered genetic variants in AAV, the aim of this study was to investigate whether these genetic variants are false-positives or are truly associated with AAV by applying a Bayesian approach. Moreover, we discussed and re-analyzed the filtered data based on the integration of the available results for genetic variants.

## 2. Experimental Section

### 2.1. Method

#### Database Search and Selection

The eligible studies were selected according to the standardized reporting protocol of systematic reviews and meta-analyses PRISMA (Preferred Reporting Items for Systematic reviews and Meta-Analyses) checklist (Appendix A) [7]. A systematic search was performed in PubMed and EMBASE to retrieve studies published before 20 February 2018. Antineutrophil cytoplasmic antibody (ANCA)-associated vasculitis, polymorphisms, meta, genetic, variant and genome-wide association study (GWAS) were the terms used in the systematic search. At the end of the systematic search, 241 articles were identified, 13 were screened in detail, and 7 were selected for analysis [4,5,6,8,9,10,11]. (Figure 1 and Appendix A).

The inclusion criteria were (1) a genetic meta-analysis or GWAS providing information of odds ratios (OR), 95% confidence intervals (CI) in AAV (2) studies written in English, and (3) most recent larger meta-analyses (if smaller numbers have been analyzed before). The exclusion criteria were (1) studies not related to genetic polymorphisms or AAV, (2) articles not providing accurate data sets (i.e., review articles), and (3) older meta-analyses with overlapping genetic variants.

Data elements collected from meta-analyses included, when available, author, publication year, gene variant, single nucleotide polymorphism ID (rsID), genotype comparison, OR, 95% CI, minor allelic frequency, ethnicity of study population, number of cases and controls, publication bias, and heterogeneity. Results of various subgroup analyses (i.e., Caucasian vs. mixed-population) were also extracted.

### 2.2. Statistical Method

Data related to statistically significant associations (i.e., OR, 95% CI, *p*-value) and heterogeneity parameters (i.e., *p*-value and *I*^2^) were separately extracted. Statistically significant associations were selected if the reported *p*-value was <5 × 10^−8^ for results of meta-analyses in GWASs for both the discovery and replication cohorts, and <0.05 for meta-analysis of general observational studies (i.e., case-control studies).

To assess and identify noteworthy associations, false-positive report probability (FPRP) and Bayesian false discovery probability (BFDP) were applied [12,13]. FPRP is defined as “the probability of no true association between a gene variant and disease (null hypothesis)” for a statistically significant association [12], and detailed calculation is presented in the Appendix A. FPRP is calculated with the observed p-value for the association, the statistical power of the test, and the prior probability that a molecular association is real. In this review, we calculated FPRP at two levels of assumed prior probabilities, which were 10^−3^ and 10^−6^. The interpretation for the noteworthiness of significant associations using FPRP is that the FPRP value of <0.2 is noteworthy [12].

BFDP is another Bayesian statistical method for detecting the true association between a gene variant and disease [13]. Main differences of BFDP compared to FPRP is that BFDP is independent from a statistical power and its approximation is based on a logistic regression model instead of a standard normal distribution [13]. The interpretation for the noteworthiness of significant associations using BFDP is that the BFDP value of <0.8 is noteworthy [13].

### 2.3. Outcomes

This study presents both FPRP and BFDP methods because the genetic epidemiologists and clinicians use BFDP less frequently than FPRP [12], although BFDP is a more recently developed method with a more substantial justification for its use [13]. By summarizing both results of FPRP and BFDP, we provide the readers options for interpreting noteworthiness.

### 2.4. Construction of PPI (Protein–Protein Interaction) Network

The STRING 9.1 network database is one of the largest databases of direct (physical) protein–protein interactions and indirect (functional) interactions constructed from various data sources including genomic context predictions, high-throughput experiments, co-expression, and known databases [14]. The STRING database covers 9.6 million proteins from more than 2031 organisms. In our study, we used STRING database to identify the PPIs associated with genes mapping to AAV, GPA, PR3-ANCA, and MPO-ANCA SNPs.

## 3. Results

### 3.1. Computation of Noteworthy Variants

The re-analysis of genetic variants was mainly driven by two categories of studies: one meta-analysis of observational studies and two GWASs with replication cohorts (combined analysis). All statistically significant SNPs and variants (*p* < 0.05 for observational studies and *p* < 5.0 × 10^−8^ for GWASs) reported in the meta-analyses were included in this study. In addition, variants found in the GWAS meta-analysis with 5.0 × 10^−8^ < *p* < 0.05, which were rejected as insignificant in their interpretation, were re-analyzed. Whether a variant was noteworthy was determined based on satisfaction of the condition in at least one of the values (FPRP < 0.2 and BFDP < 0.8). The reported results were based on the various subtypes (GPA and MPA). In this study we followed the classification system used in the original studies.

First, only 42/158 (27%) genetic comparisons in the meta-analysis of observational studies were noteworthy after re-analysis. In AAV, FPRP estimation resulted in 8/42 (19%) and 3/42 (7%) noteworthy genetic comparisons at the prior probability of 10^−3^ and 10^−6^ with the statistical power to detect an OR of 1.2. Likewise, 16/42 (38%) and 12/42 (26%) of comparisons were noteworthy at the prior probability of 10^−3^ and 10^−6^ with the statistical power to detect an OR of 1.5. BFDP estimation demonstrated 10/42 (24%) and 6/42 (14%) comparisons noteworthy at the prior probability of 10^−3^ and 10^−6^, respectively (Table 1 and Appendix A). 

In GPA, 4/50 (8%) and 1/50 (2%) comparisons were noteworthy in FPRP with the statistical power to detect an OR of 1.2 at the prior probability of 10^−3^ and 10^−6^; 9/50 (18%), 3/50 (6%) of comparisons with the statistical power to detect an OR of 1.5 at the prior probability of 10^-3^ and 10^-6^, respectively. In comparison, 15/50 (30%) and 9/50 (18%) variations were noteworthy in BFDP estimation at the prior probability of 10^−3^ and 10^−6^ (Table 1 and Appendix A).

In PR3-ANCA, 5/24 (21%) comparisons were noteworthy in FPRP while 10/24 (42%) were found noteworthy in BFDP. Variants marked with ‘NA’ in the tables were not assessable with FPRP due to the mathematical error of calculating the inverse of the cumulative distribution, but BFDP was still computable. All the noteworthy variants in PR3-ANCA with FPRP estimation were also noteworthy in BFDP estimation (Table 2 and Appendix A). No noteworthy comparison was observed in both MPA and MPO-ANCA vasculitis (Table 1 and Table 2 and Appendix A).

Secondly, 22/28 (79%) SNPs in the meta-analyses of GWASs were noteworthy in total. Without exception, statistically significant SNPs discovered by GWAS meta-analyses (*p* < 5.0 × 10^−8^) were also noteworthy in FPRP estimation at the prior probabilities of either 10^-3^ or 10^-6^ with the statistical power to detect an OR of 1.2 or 1.5 and in BFDP estimation at the probability of 10^−3^ or 10^−6^ except where mathematical errors did not allow FPRP. However, among the 14 SNPs with *p*-values ranging between 0.05 and 5.0 × 10^−8^, which were considered as statistically non-significant in GWAS meta-analyses, 8 SNPs were reported noteworthy in both FPRP and BFDP estimation (Table 3 and Appendix A).

We could re-analyze one GWAS of patients with GPA, while the remaining two GWASs were unable to be re-assessed due to the absence of 95% CI data. With our re-analyses, all the significant SNPs (*p* < 5.0 × 10^8^) observed in the GWASs were noteworthy in FPRP and BFDP estimation. However, among the borderline SNPs rejected by the GWAS with *p* > 5.0 × 10^−8^, 53/62 (85%) of the SNPs were found noteworthy in our re-analysis (Table 4 and Appendix A). 

### 3.2. Gene Network Analysis

FPRP and BFDP computation reported all the SNPs with genome-wide significance discovered by GWASs as noteworthy, which may indicate that SNPs identified to be significant within GWASs are highly credible due to a conservative statistical standard and larger sample size. Based on this result, the SNPs’ related protein–protein interaction (PPI) network of AAV subgroups and a Venn diagram of SNPs were constructed with the noteworthy genetic variants sorted by FPRP and BFDP and all the meaningful GWAS SNPs. The Venn diagram showed that no SNPs overlap between PR3-ANCA and MPO-ANCA vasculitis (Figure 2A and Appendix A). The PPI networks among genes mapping to PR3-ANCA and MPO-ANCA vasculitis-associated SNPs showed closer connections (Figure 2B,C). Interestingly, the genes associated with interferon-gamma (IFN-γ)-mediated signaling pathway are highly enriched in both PR3-ANCA and MPO-ANCA vasculitis-associated SNPs (e.g., yellow nodes in the PPI networks). In addition, the MPA subtype was not included in the diagram, as no noteworthy SNPs have been reported in our analysis. The PPI network of AAV showed a variety of network connections. The major genes included in the network were major histocompatibility complex (MHC) class I and II genes (i.e., human leukocyte antigen (HLA)-A, -B, -C, -G, -F, -DPA1, -DPB1, -DQB1, -DRA). Other subtypes are shown in Figure 3 and Appendix A.

To investigate disease pathways, gene ontology (GO) enrichment analysis was performed. Several GO terms enriched in the SNPs set were identified by GO enrichment analysis for each subtype. Immune-related GO terms, including “antigen processing and presentation of peptide or polysaccharide antigen via MHC class II” (GO:0002504) with a *p*-value of 4.4 × 10^−17^, “interferon-gamma-mediated (IFN-γ) signaling pathway” (GO:0060333) with a *p*-value of 2.9 × 10^−12^, and “T cell receptor signaling pathway” (GO:0050852) with a *p*-value of 5.5 × 10^−10^, were identified as the most prominently enriched AAV-related gene sets (Figure 4 and Appendix A)

### 3.3. Bullets

Several studies in ANCA-associated vasculitis have reported statistically significant genetic variants.Integration and further analysis of the data from several studies using FPRP, BFDP, and GO analysis identified novel biological pathways in ANCA-associated vasculitis, including a role of the IFN-γ pathway.Our results underline that there is a dichotomy between the respective target antigens, PR3 and MPO.

## 4. Discussion

In this study we investigated whether the genetic variants previously found to be associated with AAV are truly associated with AAV or are false-positives. Former studies have demonstrated that the target antigens of AAV, MPO, and PR3 are better discriminators of the disease than the clinical phenotypes, MPA and GPA [2]. Meta-analyses and GWASs have confirmed a genetic background of AAV. Our analysis found no genetic overlap of MPO-ANCA and PR3-ANCA vasculitis. This further underlines the need to distinguish both subtypes in clinical studies and may lead to a re-classification of AAV based on the serotypes. Our analyses further highlight that AAV is not a multigenic disease with the implication of several genetic variants contributing to one’s individual risk to develop vasculitis.

This work used FPRP and BFDP estimation to re-analyze genetic associations from the retrieved research papers. Discovered SNPs of each included study with *p*-values under 0.05 or either 5.0 × 10^−8^ were computed by both methods. Nearly half of the significant SNPs reported in meta-analyses of the observational studies were not noteworthy, whereas those SNPs in GWASs and their meta-analyses were noteworthy with FPRP and BFDP. However, when we applied the Bayesian procedures to SNPs associated with borderline error probabilities (5.0 × 10^−8^ < *p*-value < 0.05) observed in a GWAS including patients with GPA, FPRP and BFDP computation yielded a noteworthiness of 85% of these SNPs.

The exact mechanisms leading to AAV onset are unclear, but a genetic predisposition and self-intolerance to environmental exposures, such as infections, have been proposed [14]. A current pathogenetic model proposes the initiation of an inflammatory cascade with the release of PR3 (pro-inflammatory cytokine) by an unknown insult, further selecting PR3-specific B lymphocytes producing ANCA and PR3 specific T cells [14]. Subsequently, neutrophils are activated by ANCA priming, leading to the destruction of endothelial cells by firmly attached neutrophils on endothelial cells [15,16]. Our GO enrichment analysis showed that the processing of antigens via MHC class II, the IFN-γ-mediated pathway and the T cell receptor signaling pathway are the main biological processes disturbed in AAV. GO enrichment analysis identified a crucial role of the IFN-γ mediated signaling pathway and showed comparable involvement in both, MPO-ANCA and PR3-ANCA vasculitis. PPI analysis revealed a role of the interferon regulatory factor 5 (IRF5) in the model including patients with GPA and AAV. In systemic lupus erythematosus (SLE), IRF5 genetic variants are associated with an increased risk to develop the disease, alongside elevation of IRF5 expression and IFN production [17]. IFN-γ has been studied in two biomarker studies involving a cohort with AAV and one cohort of patients with GPA. Analysis of samples obtained from patients with AAV recruited to participate in the RAVE trial indicated significant higher levels of IFN-γ during active disease compared to remission (137 samples each) and to healthy controls (68 samples) [18]. Furthermore, the expression of chemokine (C-X-C motif) ligand (CXCL10 or IFN-γ-induced protein 10) was tested in the same study. CXCL10 expression is dependent on IFN-γ and is implicated in several autoimmune disorders [19]. In patients with AAV, CXCL10 levels did not differ among patients with active disease or in remission, while CXCL10 levels were significantly lower in healthy controls. This argues for a sufficient reduction of IFN-γ once remission is achieved in AAV, but in contrast, the levels of CXCL10 remain elevated. More research is clearly needed to understand the regulation of the IFN-γ pathway in AAV.

IFN-γ increases MHC class II expression in antigen-presenting cells (B cells, macrophages, and dendritic cells) that are engaged by microbial invasions [20,21]. Furthermore, IFN-γ negatively regulates anti-inflammatory IL-10 production and induces pro-inflammatory IL-12 secretion [22]. IL-12 drives helper T(Th) cells to differentiate towards a Th1 phenotype and stimulates natural killer (NK) cells to produce IFN-γ. IFN-γ secreted from NK cells induces monocyte differentiation towards active macrophages in the local inflammatory site [23]. In addition, MPO stimulation leads to a dose-dependent production of IFN-γ by monocytes [24]. MPO induces the formation of neutrophil extracellular traps (NET), which is thought to be a crucial factor in the cascade of inflammation in AAV [25]. The IFN-γ pathway may play a role in the development of local inflammation. Alongside the immune system activation and NET formation as crucial mechanisms in the inflammatory processes, our data confirm that genetic predisposition and the activation of antigen-presenting cells with the selection of autoimmune T cells are important in the pathogenesis of AAV.

Our PPI network shows the mainframe structure of genetic interactions among HLA-associated genes and COL11A2 in GPA, PR3-ANCA vasculitis, and AAV itself. COL11A2 gene codes for the pro-α2 chain of collagen type XI. COL11A2 was demonstrated to be in linkage disequilibrium with the HLA-DP gene in an AAV GWAS [3,26]. Except MHC coding genes, other genetic risk factors associated with AAV were PTPN22, RXRB, CTLA-4, MICA, SERPINA1, PRTN3, CD226, TLR9, IRF5, NOTCH4, AGER, and CFB, which is consistent with previous findings (Figure 3). PTPN22 encodes protein tyrosine phosphatase in lymphoid tissues with an abnormal regulatory CD4 T-cell (Treg) function and increased neutrophil function reported in PTPN22 variant (rs2476601) [27]. RXRB is a gene encoding a family of retinoid X receptors, which form homodimers and heterodimers with retinoic acid, thyroid hormone, and vitamin D receptors [28]. There was a functional difference in the responsiveness to vitamin A and vitamin D in AAV patients, which could not be explained by RXRB polymorphisms alone [29]. CTLA-4 is a gene encoding an inhibitory surface protein on activated T cells that interacts with CD80 or CD86 and competes with CD28 (co-stimulatory molecule) [30]. Abatacept, a selective modulator of the CD80/86-CD28 costimulatory signal was successfully used in patients with non-severe GPA and is currently in a phase III trial (ABROGATE, NCT02108860), which suggests that the CTLA-4 protein might be of importance in AAV pathogenesis [31]. SERPINA1 and PRTN3 are also referred to as crucial genes in the pathogenesis of PR3-ANCA vasculitis [27]. PRTN3 is the gene encoding PR3 that exists on the surface or in the cytoplasm of neutrophils, whereas SERPINA1 encodes α-1 antitrypsin, the major inhibitory molecule of PR3. Previous findings suggest that PRTN3 genetic variants determine the predominant location of PR3 in neutrophils, where it may affect the activity of neutrophils [27]. No noteworthy variants were observed in MPA, but genetic variants related to the HLA-DR and HLA-DQ loci were found to be associated with MPO-ANCA vasculitis (Appendix A).

FPRP was criticized with respect to its heuristic derivation of the formula, especially with the use of α and 1-β as the probabilities of observing values greater or less than the test quantities under a null and alternative hypothesis [32]. Even though a labored statistical derivation exists, the concept of “noteworthiness” originated from FPRP has influenced recent genetic studies [12]. Moreover, although direct comparison between BFDP and FPRP is not possible, the behavior of both is similar in promoting SNPs’ rankings [13]. BFDP produced more noteworthy findings than FPRP because the latter derives smaller posterior null estimates due to the conditioning on tail areas unlike the reliance on point estimates provided by BFDP [13,33].

Our study has some limitations. First, different GWASs included in this study share the same cohorts. Therefore, slight overestimation of some sharing genes found in these studies may exist. However, no effect will be present in the gene network analysis because weighting of a gene corresponded to the number of significant SNPs in each gene. Second, we were not able to analyze variants with incomplete data (those without 95% CIs). Thus, it was not possible to confirm that all genome-wide significant SNPs satisfied the FPRP and BFDP thresholds. In fact, SNPs significant in GWASs were included in GO enrichment analysis, suggesting that these SNPs would be noteworthy due to previous results (Table 3 and Table 4 and Appendix A). Furthermore, the low frequency of patients with MPA in the included studies may have led to the finding of no noteworthy SNPs in our re-analysis. Lastly, linkage disequilibrium and expression changes of SNPs were not considered in this analysis, perhaps treating SNPs and its properties too simply. Further research comprising quantitative expression of loci is needed.

Nevertheless, our study merged current results of genetic associations in AAV and its subtypes. In addition, the investigation of false-positive results in genetic research proves that researchers should pay careful attention when interpreting the positive results reported in previous papers. All GWAS SNPs associated with a borderline *p*-value are worth further examination with various statistical methods, as we found many of them to be noteworthy with the used Bayesian methods. Moreover, we highlight the importance of the IFN-γ pathway in the pathogenesis of AAV through GO enrichment analysis using GWAS and meta-analysis gene sets. This suggests that further research into the IFN-γ pathway in AAV may lead to the development of novel therapeutic approaches for this complex disease.

## Figures and Tables

**Figure 1 jcm-08-00266-f001:**
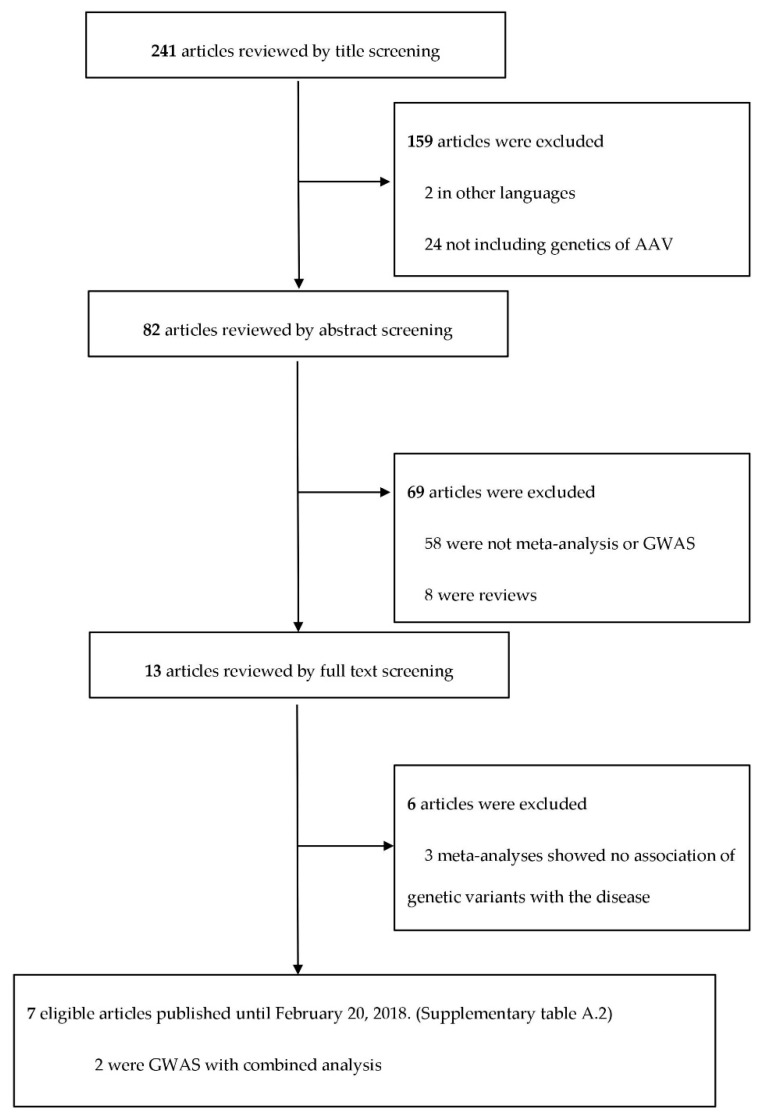
The process of the systematic search performed to study genetic variants in ANCA-associated vasculitis. Abbreviations used: AAV (ANCA-associated vasculitis), ANCA (anti-neutrophil cytoplasmic antibody), and GWAS (genome-wide association study).

**Figure 2 jcm-08-00266-f002:**
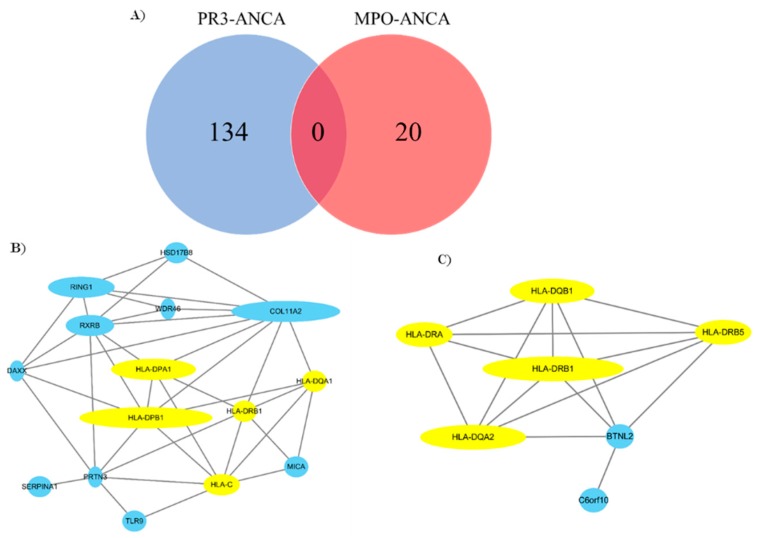
The number of noteworthy SNPs in PR3-ANCA and MPO-ANCA and their protein–protein networks (PPIs). (**A**) The overlap of SNPs in PR3-ANCA and MPO-ANCA; the lack of any shared SNPs between these two subtypes of ANCA-associated vasculitis. (**B**) PPI network of genes associated with PR3-ANCA vasculitis. (**C**) PPI network of genes associated with MPO-ANCA vasculitis. Borderline SNPs (5.0 × 10^−8^ < *p* < 0.05) from GWASs that have been noteworthy in our analysis were also included. The yellow nodes represent the genes associated with the interferon-gamma-mediated signaling pathway. The number of SNPs associated with PR3-ANCA and MPO-ANCA is 134 and 20, respectively, which leads to a total of 21 and 8 unique genes. Note that some genes have multiple SNPs associated with PR3-ANCA and MPO-ANCA. Among these genes, only 15 genes and 7 genes are included in the PPI network by revealing known PPI interactions in Figure 2B,C.

**Figure 3 jcm-08-00266-f003:**
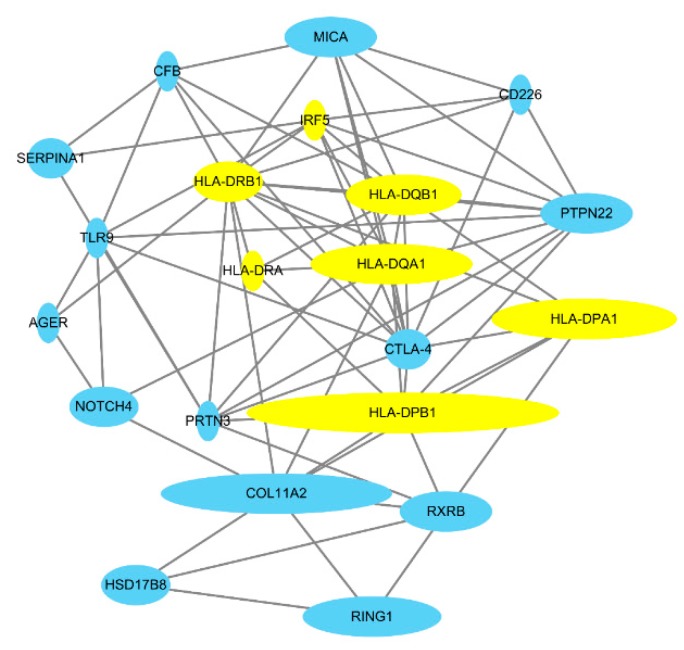
Protein–protein interaction network of associated genes in the etiopathogenesis of ANCA-associated vasculitis. The size of the node indicates the number of noteworthy SNPs of each gene. This network includes noteworthy SNPs and GWAS SNPs. The variants are listed in Appendix A. The yellow nodes represent the genes associated with the interferon-gamma-mediated signaling pathway.

**Figure 4 jcm-08-00266-f004:**
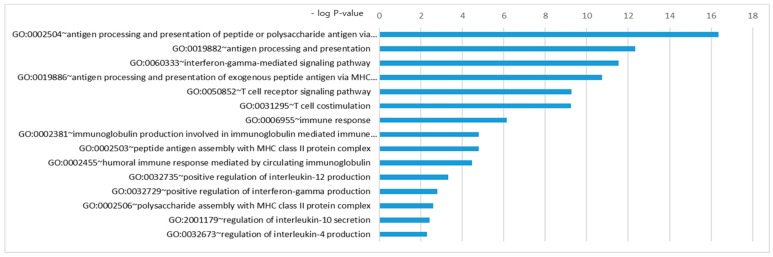
Gene ontology enrichment analysis of ANCA-associated vasculitis. The analysis was performed with the associated genetic variants. The cut-off p-value for this analysis was 0.01 (−log *p*-value = 2). Other subtype results are shown in the Appendix A.

**Table 1 jcm-08-00266-t001:** Meta-analysis results of observational studies; gene variants with statistical significance (*p*-value < 0.05), found to be noteworthy by false-positive report probability (FPRP) or Bayesian false discovery probability (BFDP) for each clinical diagnosis antineutrophil cytoplasmic antibody (ANCA)-associated vasculitis (AAV), granulomatosis with polyangiitis (GPA), and microscopic polyangiitis (MPA)).

Gene/Variant	Minor Allele/Comparison	OR (95% CI)	*p*-Value for Meta-Analysis	Publications (*n*)	Diagnosis (Clinical Subtypes)	No. of Cases/Controls	*I*^2^ (%)	*I*^2^ (P)	Egger’s *p*-Value	Power OR 1.2	Power OR 1.5	FPRP Values at Prior Probability				BFDP 0.001	BFDP 0.000001	Author, Year
												OR 1.2		OR 1.5				
												0.001	0.000001	0.001	0.000001			
CD226 rs763361	T	1.14 (1.07–1.21)	<0.001	3	AAV	2422/17898	0	0.444	0.792	0.954	1.000	0.017	0.945	0.016	0.942	0.437	0.999	Rahmattulla, et al. 2016 [5]
CTLA-4 rs3087243 (CT60)	A	0.81 (0.75–0.87)	<0.001	3	AAV	2015/7855	25	0.262	0.122	0.218	1.000	0.000	0.033	0.000	0.007	0.001	0.347	Rahmattulla, et al. 2016 [5]
CTLA-4	(AT)_86_	0.54 (0.43–0.67)	<0.001	4	AAV	303/543	89	<0.001	0.946	0.000	0.028	0.348	0.998	0.001	0.437	0.010	0.913	Rahmattulla, et al. 2016 [5]
CTLA-4 rs3087243 (CT60)	AA vs. GG	0.693 (0.512–0.796)	6.39 × 10^−5^	2	AAV	*797/9669*	57.7	0.124		0.005	0.708	0.045	0.824	0.000	0.029	0.018	0.948	Lee, et al. 2012 [8]
HLA-DPA1 rs9277341	C	0.35 (0.30–0.40)	<0.001	2	AAV	1032/2200	54	0.116	0.215	NA	NA	NA	NA	NA	NA	0.000	0.000	Rahmattulla, et al. 2016 [5]
HLA-DPB2 rs3130215	A	1.40 (1.29–1.52)	<0.001	3	AAV	1417/7249	99	<0.001	0.446	0.000	0.950	0.000	0.000	0.000	0.000	0.000	0.000	Rahmattulla, et al. 2016 [5]
HLA-DRB4	-	1.69 (1.36–2.10)	<0.001	4	AAV	260/1845	61	0.055	0.533	0.001	0.141	0.686	1.000	0.015	0.940	0.259	0.997	Rahmattulla, et al. 2016 [5]
HSD17B8 rs421446	C	0.40 (0.34–0.48)	<0.001	2	AAV	738/1872	0	0.620	NA	NA	NA	NA	NA	NA	NA	0.000	0.000	Rahmattulla, et al. 2016 [5]
IRF5 rs10954213	G	0.77 (0.70–0.83)	<0.001	3	AAV	1535/6977	99	<0.001	0.948	0.019	1.000	0.000	0.000	0.000	0.000	0.000	0.001	Rahmattulla, et al. 2016 [5]
PTPN22 rs2476601	T vs. C	1.415 (1.228–1.630)	1.59 × 10^−6^	3	AAV	*1184/10459*	0	0.393	0.481	0.011	0.791	0.119	0.931	0.002	0.160	0.091	0.990	Lee, et al. 2012 [8]
PTPN22 rs2476601	A	1.39 (1.24–1.56)	<0.001	4	AAV	2099/8678	0	0.693	0.500	0.006	0.902	0.004	0.780	0.000	0.024	0.002	0.654	Rahmattulla, et al. 2016 [5]
RING1/RXRB rs213213	A	1.71 (1.57–1.86)	<0.001	3	AAV	1414/7238	73	0.026	0.187	NA	NA	NA	NA	NA	NA	0.000	0.000	Rahmattulla, et al. 2016 [5]
RXRB rs6531	C	1.63 (1.50–1.77)	<0.001	3	AAV	1557/6955	96	<0.001	0.292	NA	NA	NA	NA	NA	NA	0.000	0.000	Rahmattulla, et al. 2016 [5]
RXRB rs9277935	T	0.44 (0.37–0.50)	<0.001	3	AAV	1417/7233	73	0.025	0.393	NA	NA	NA	NA	NA	NA	0.000	0.000	Rahmattulla, et al. 2016 [5]
SERPINA 1	Z allele	2.94 (2.22–3.88)	<0.001	8	AAV	3662/8581	41	0.092	0.078	0.000	0.000	0.173	0.995	0.000	0.025	0.000	0.005	Rahmattulla, et al. 2016 [5]
TLR9 rs352162	T	1.58 (1.43–1.75)	<0.001	1	AAV	1289/1898	96	<0.001	NA	NA	NA	NA	NA	NA	NA	0.000	0.000	Rahmattulla, et al. 2016 [5]
PTPN22 rs2476601	T vs. C	2.042 (1.534–2.719)	1.02 × 10^−6^	2	ANCA (+) GPA	*-*	0	0.989	NA	0.000	0.017	0.882	0.999	0.056	0.855	0.375	0.998	Lee, et al. 2012 [8]
CD226 rs763361	T	1.19 (1.11–1.28)	<0.001	3	GPA	*2021/17898*	72.2	0.006		0.589	1.000	0.005	0.832	0.003	0.745	0.124	0.993	Rahmattulla, et al. 2016 [5]
CLTA-4 rs3087243	A	0.80 (0.73–0.87)	<0.001	3	GPA	*1561/7855*	38.7	0.180		0.170	1.000	0.001	0.521	0.000	0.156	0.011	0.915	Rahmattulla, et al. 2016 [5]
CTLA-4	(AT)_86_	0.44 (0.34–0.57)	<0.001	3	GPA	*210/432*	86.5	0.001		0.000	0.001	0.434	0.999	0.001	0.381	0.002	0.670	Rahmattulla, et al. 2016 [5]
CTLA-4 rs3087243 (CT60)	A vs. G	0.79 (0.70–0.89)	9.83 × 10^−5^	2	GPA	*880/1969*				0.190	0.997	0.358	0.982	0.096	0.914	0.779	1.000	Chung, et al. 2012 [10]
HLA-DPA1 rs9277341	C	0.35 (0.30–0.41)	<0.001	2	GPA	*1032/2200*	54.8	0.109		NA	NA	NA	NA	NA	NA	0.000	0.000	Rahmattulla, et al. 2016 [5]
HLA-DPB1*0301	-	0.23 (0.16–0.32)	<0.001	3	GPA	*774/918*	61.7	0.050		NA	NA	NA	NA	NA	NA	0.000	0.000	Rahmattulla, et al. 2016 [5]
HLA-DPB1*0401	-	2.89 (2.50–3.35)	<0.001	3	GPA	*774/918*	67.5	0.026		NA	NA	NA	NA	NA	NA	0.000	0.000	Rahmattulla, et al. 2016 [5]
HLA-DR6	-	0.45 (0.33–0.62)	<0.001	4	GPA	*301/6132*	59.8	0.058		0.000	0.008	0.927	1.000	0.114	0.992	0.511	0.999	Rahmattulla, et al. 2016 [5]
IRF5 rs10954213	G	0.66 (0.59–0.74)	<0.001	2	GPA	*1021/6267*	99.1	0.000		0.000	0.432	0.000	0.033	0.000	0.000	0.000	0.000	Rahmattulla, et al. 2016 [5]
PTPN22 rs2476601	A	1.43 (1.26–1.62)	<0.001	4	GPA	*1616/8678*	0.0	0.411		0.003	0.774	0.006	0.867	0.000	0.024	0.002	0.649	Rahmattulla, et al. 2016 [5]
RING1/RXRB rs213213	A	1.91 (1.73–2.10)	<0.001	3	GPA	*1132/7238*	0.0	0.551		NA	NA	NA	NA	NA	NA	0.000	0.000	Rahmattulla, et al. 2016 [5]
RXRB rs6531	C	1.70 (1.55–1.86)	<0.001	3	GPA	*1211/6955*	96.5	0.000		NA	NA	NA	NA	NA	NA	0.000	0.000	Rahmattulla, et al. 2016 [5]
RXRB rs9277935	T	0.37 (0.31–0.43)	<0.001	3	GPA	*1135/7233*	0.0	0.798		NA	NA	NA	NA	NA	NA	0.000	0.000	Rahmattulla, et al. 2016 [5]
SERPINA 1	Z allele	2.40 (1.73–3.33)	<0.001	4	GPA	*972/2636*	0.0	0.763		0.000	0.002	0.906	1.000	0.062	0.985	0.282	0.997	Rahmattulla, et al. 2016 [5]
HLA-DRB4	-	2.06 (1.57–2.69)	<0.001	2	EGPA	*150/691*	0.4	0.316		0.000	0.010	0.754	1.000	0.011	0.918	0.089	0.990	Rahmattulla, et al. 2016 [5]

FPRP: false-positive report probability; BFDP: Bayesian false discovery probability; OR: odds ratio; 95% CI: 95% confidence interval; No.: number; NA: not applicable.

**Table 2 jcm-08-00266-t002:** Meta-analysis results of observational studies; gene variants with statistical significance (*p*-value < 0.05), found to be noteworthy by FPRP or BFDP for each serologic diagnosis (MPO-ANCA and PR3-ANCA).

Gene/Variant	Minor Allele/ Comparison	OR (95% CI)	*p*-Value for Meta-Analysis	Publications (*n*)	Diagnosis (Serologic Subtypes)	No. of Cases/Controls	I^2^ (%)	I^2^ (P)	Egger’s *p*-Value	Power OR 1.2	Power OR 1.5	FPRP Values at Prior Probability				BFDP 0.001	BFDP 0.000001	Author, Year
												OR 1.2		OR 1.5				
												0.001	0.000001	0.001	0.000001			
HLA-DPA1 rs9277341	C	0.27 (0.22–0.33)	<0.001	1	PR3-ANCA	578/1820	-	-		NA	NA	**NA**	**NA**	**NA**	**NA**	**0.000**	**0.000**	Rahmattulla, et al. 2016 [5]
HLA-DPB1*0401	-	3.93 (2.75–5.62)	<0.001	2	PR3-ANCA	183/139	0.0	0.960		0.000	0.000	0.615	0.999	**0.001**	0.495	**0.000**	**0.170**	Rahmattulla, et al. 2016 [5]
HLA-DPB2 rs3130215	A	0.65 (0.55–0.77)	<0.001	1	PR3-ANCA	326/5366	-	-		0.002	0.385	0.235	0.997	**0.002**	0.618	**0.062**	0.985	Rahmattulla, et al. 2016 [5]
HLA-DRB1*15	-	2.82 (2.00–3.96)	<0.001	2	PR3-ANCA	131/582	84.0	0.002		0.000	0.000	0.842	1.000	**0.016**	0.942	**0.040**	0.977	Rahmattulla, et al. 2016 [5]
RING1/RXRB rs213213	A	2.06 (1.75–2.41)	<0.001	1	PR3-ANCA	326/5366	-	-		NA	NA	**NA**	**NA**	**NA**	**NA**	**0.000**	**0.000**	Rahmattulla, et al. 2016 [5]
RXRB rs6531	C	2.19 (1.92–2.51)	<0.001	1	PR3-ANCA	478/5251	-	-		NA	NA	**NA**	**NA**	**NA**	**NA**	**0.000**	**0.000**	Rahmattulla, et al. 2016 [5]
RXRB rs9277935	T	0.24 (0.17–0.33)	<0.001	1	PR3-ANCA	326/5350	-	-		NA	NA	**NA**	**NA**	**NA**	**NA**	**0.000**	**0.000**	Rahmattulla, et al. 2016 [5]
SERPINA 1	Z allele	3.53 (2.28–5.49)	<0.001	5	PR3-ANCA	280/4788	21.3	0.279		0.000	0.000	0.963	1.000	0.229	0.997	**0.512**	0.999	Rahmattulla, et al. 2016 [5]
TLR9 rs352140	T	1.28 (1.12–1.45)	0.018	1	PR3-ANCA	NR/NR	0.0	0.782		0.155	0.994	0.402	0.999	**0.095**	0.991	**0.778**	1.000	Rahmattulla, et al. 2016 [5]
TLR9 rs352162	T	1.30 (1.14–1.47)	<0.001	1	PR3-ANCA	NR/NR	0.0	0.503		0.101	0.989	0.221	0.996	**0.028**	0.967	**0.532**	0.999	Rahmattulla, et al. 2016 [5]

**Table 3 jcm-08-00266-t003:** Results of meta analyses (combined analysis) with genome-wide association studies and replication cohort. Noteworthy genetic variants with satisfied FPRP or BFDP values are presented. SNPs: single nucleotide polymorphism.

Gene/Variant	Comparison	OR (95% CI)	*p*-Value	Diagnosis (Clinical/Serologic Subtypes)	Ethnicity	No. of Cases/Controls	Power OR 1.2	Power OR 1.5	FPRP Values at Prior Probability				BFDP 0.001	BFDP 0.000001	Author, Year
									OR 1.2		OR 1.5				
									0.001	0.000001	0.001	0.000001			
**SNPs Statistically Significant (*p* < 5.00 × 10^−8^)**															
HLA-DPA1 rs9277341	T vs. C	2.44 (2.21–2.69)	6.09 × 10^−71^	AAV	Caucasian	1986/4723	NA	NA	NA	NA	NA	NA	0.000	0.000	Merkel, et al. 2017 [4]
HLA-DPB1 rs1042169	G vs. A	2.82 (2.54–3.13)	1.12 × 10^−84^	AAV	Caucasian	1986/4723	NA	NA	NA	NA	NA	NA	0.000	0.000	Merkel, et al. 2017 [4]
HLA-DPB1 rs141530233	A del	2.99 (2.69–3.33)	1.13 × 10^−89^	AAV	Caucasian	1986/4723	NA	NA	NA	NA	NA	NA	0.000	0.000	Merkel, et al. 2017 [4]
HLA-DQA1 rs35242582	A vs. G	1.60 (1.46–1.76)	6.34 × 10^−23^	AAV	Caucasian	1986/4723	NA	NA	NA	NA	NA	NA	0.000	0.000	Merkel, et al. 2017 [4]
HLA-DQB1 rs1049072	A vs. G	1.40 (1.28–1.53)	6.46 × 10^−13^	AAV	Caucasian	1986/4723	0.000	0.936	0.000	0.000	0.000	0.000	0.000	0.000	Merkel, et al. 2017 [4]
PRTN3 rs62132293	G vs. C	1.29 (1.19–1.39)	8.60 × 10^−11^	AAV	Caucasian	1986/4723	0.029	1.000	0.000	0.000	0.000	0.000	0.000	0.002	Merkel, et al. 2017 [4]
PTPN22 rs6679677	A vs. C	1.40 (1.25–1.57)	1.88 × 10^−8^	AAV	Caucasian	1986/4723	0.004	0.881	0.002	0.172	0.000	0.001	0.001	0.447	Merkel, et al. 2017 [4]
SERPINA1 rs28929474	T vs. C	2.18 (1.75–2.71)	3.09 × 10^−12^	AAV	Caucasian	1986/4723	0.000	0.000	0.056	0.855	0.000	0.001	0.000	0.010	Merkel, et al. 2017 [4]
HLA-DPA1 rs9277341	C vs. T	0.33 (0.28–0.39)	2.18 × 10^−39^	GPA	Caucasian	750/1820	NA	NA	NA	NA	NA	NA	0.000	0.000	Xie, et al. 2013 [6]
HLA-DPB1 rs9277554	T vs. C	0.24 (0.20–0.30)	1.92 × 10^−50^	GPA	Caucasian	750/1820	NA	NA	NA	NA	NA	NA	0.000	0.000	Xie, et al. 2013 [6]
SEMA6A rs26595	C vs. T	0.74 (0.67–0.82)	2.09 × 10^−8^	GPA	Caucasian	987/2731	0.012	0.977	0.001	0.071	0.000	0.001	0.001	0.423	Xie, et al. 2013 [6]
HLA-DQA2 rs3998159	C vs. A	2.72 (2.24–3.22)	5.24 × 10^−25^	MPO-ANCA	Caucasian	378/4723	NA	NA	NA	NA	NA	NA	0.000	0.000	Merkel, et al. 2017 [4]
HLA-DQA2 rs7454108	C vs. T	2.73 (2.25–3.24)	5.03 × 10^−25^	MPO-ANCA	Caucasian	378/4723	NA	NA	NA	NA	NA	NA	0.000	0.000	Merkel, et al. 2017 [4]
HLA-DQB1 rs1049072	A vs. G	2.37 (2.01–2,78)	2.13 × 10^−24^	MPO-ANCA	Caucasian	378/4723	NA	NA	NA	NA	NA	NA	0.000	0.000	Merkel, et al. 2017 [4]
**SNPs with Statistically Borderline Significance (5.00 × 10^−8^ ≤ *p* < 0.05)**															
PTPN22(R620W) rs2476601	A vs. G	1.36 (1.21–1.53)	1.86 × 10^−7^	AAV	Caucasian	1986/4723	0.019	0.948	0.016	0.625	0.000	0.032	0.020	0.953	Merkel, et al. 2017 [4]
CCDC86 rs595018	A vs. G	1.46 (1.27–1.69)	1.60 × 10^−7^	GPA	Caucasian	1986/4723	0.004	0.641	0.084	0.902	0.001	0.058	0.033	0.971	Xie, et al. 2013 [6]
COBL rs1949829	T vs. C	1.78 (1.42–2.24)	4.19 × 10^−7^	GPA	Caucasian	1986/4723	0.000	0.072	0.694	0.996	0.012	0.549	0.177	0.995	Xie, et al. 2013 [6]
DCTD rs4862110	C vs. T	1.44 (1.24–1.67)	2.14 × 10^−6^	GPA	Caucasian	1986/4723	0.008	0.705	0.151	0.947	0.002	0.167	0.092	0.990	Xie, et al. 2013 [6]
DOK4 rs6023640	T vs. G	1.29 (1.14–1.45)	2.73 × 10^−5^	GPA	Caucasian	987/2731	0.113	0.994	0.148	0.946	0.019	0.664	0.445	0.999	Xie, et al. 2013 [6]
FLJ34870 rs7585252	G vs. A	1.26 (1.13–1.40)	1.74 × 10^−5^	GPA	Caucasian	987/2731	0.182	0.999	0.086	0.904	0.017	0.632	0.408	0.999	Xie, et al. 2013 [6]
PAEP rs705669	G vs. A	0.77 (0.68–0.87)	2.52 × 10^−5^	GPA	Caucasian	987/2731	0.102	0.990	0.210	0.964	0.027	0.733	0.520	0.999	Xie, et al. 2013 [6]
WSCD1 rs7503953	A vs. C	1.50 (1.29–1.76)	1.93 × 10^−7^	GPA	Caucasian	1986/4723	0.003	0.500	0.176	0.955	0.001	0.117	0.058	0.984	Xie, et al. 2013 [6]

**Table 4 jcm-08-00266-t004:** Re-analysis of the SNPs discovered in genome-wide association studies of patients with GPA. Noteworthy genetic variants with satisfied FPRP or BFDP values are presented. Xie et al. only provided GPA SNPs with odds ratios (ORs) and 95% confidence intervals (CIs).

Gene/Variant	Comparison	OR (95% CI)	*p*-Value	Diagnosis (Clinical/Serologic Subtypes)	Ethnicity	No. of Cases/Controls	Power OR 1.2	Power OR 1.5	FPRP Values at Prior Probability				BFDP 0.001	BFDP 0.000001	Author, Year
									OR 1.2		OR 1.5				
									0.001	0.000001	0.001	0.000001			
**SNPs with *p*-Value < 5.00 × 10^−8^**															
HLA-DOA rs3130604	G vs. A	1.67 (1.39–2.02)	4.39 × 10^−8^	GPA	Caucasian	459/1503	0.000	0.134	0.277	0.997	**0.001**	0.487	**0.025**	0.962	Xie, et al. 2013 [6]
HLA-DOA rs763469	A vs. G	1.70 (1.41–2.04)	1.46 × 10^−8^	GPA	Caucasian	459/1503	0.000	0.089	**0.114**	0.992	**0.000**	**0.116**	**0.003**	**0.764**	Xie, et al. 2013 [6]
HLA-DPA1 rs2395309	G vs. A	0.27 (0.20–0.36)	2.15 × 10^−19^	GPA	Caucasian	459/1503	NA	NA	**NA**	**NA**	**NA**	**NA**	**0.000**	**0.000**	Xie, et al. 2013 [6]
HLA-DPA1 rs3077	C vs. T	0.27 (0.20–0.36)	2.68 × 10^−19^	GPA	Caucasian	459/1503	NA	NA	**NA**	**NA**	**NA**	**NA**	**0.000**	**0.000**	Xie, et al. 2013 [6]
HLA-DPA1 rs2301226	T vs. C	0.48 (0.36–0.62)	4.85 × 10^−8^	GPA	Caucasian	459/1503	0.000	0.006	0.613	0.999	**0.003**	0.762	**0.023**	0.959	Xie, et al. 2013 [6]
HLA-DPA1 rs9277341	C vs. T	0.30 (0.25–0.38)	1.84 × 10^−30^	GPA	Caucasian	459/1503	NA	NA	**NA**	**NA**	**NA**	**NA**	**0.000**	**0.000**	Xie, et al. 2013 [6]
HLA-DPB1 rs987870	C vs. T	0.26 (0.19–0.37)	6.09 × 10^−16^	GPA	Caucasian	459/1503	0.000	0.000	0.597	0.999	**0.001**	0.462	**0.000**	**0.155**	Xie, et al. 2013 [6]
HLA-DPB1 rs9277535	G vs. A	0.24 (0.19–0.32)	2.12 × 10^−28^	GPA	Caucasian	459/1503	NA	NA	**NA**	**NA**	**NA**	**NA**	**0.000**	**0.000**	Xie, et al. 2013 [6]
HLA-DPB1 rs9277554	T vs. C	0.22 (0.17–0.28)	4.88 × 10^−38^	GPA	Caucasian	459/1503	NA	NA	**NA**	**NA**	**NA**	**NA**	**0.000**	**0.000**	Xie, et al. 2013 [6]
HLA-DPB1 rs9277565	T vs. C	0.24 (0.18–0.32)	1.91 × 10^−24^	GPA	Caucasian	459/1503	NA	NA	**NA**	**NA**	**NA**	**NA**	**0.000**	**0.000**	Xie, et al. 2013 [6]
HLA-DPB1 rs2281389	C vs. T	0.24 (0.18–0.34)	1.69 × 10^−20^	GPA	Caucasian	459/1503	0.000	0.000	0.458	0.999	**0.000**	**0.187**	**0.000**	**0.013**	Xie, et al. 2013 [6]
HLA-DPB1 rs3128917	G vs. T	0.22 (0.17–0.29)	4.92 × 10^−33^	GPA	Caucasian	459/1503	NA	NA	**NA**	**NA**	**NA**	**NA**	**0.000**	**0.000**	Xie, et al. 2013 [6]
HLA-DPB1 rs3117222	A vs. G	0.22 (0.17–0.29)	3.05 × 10^−33^	GPA	Caucasian	459/1503	NA	NA	**NA**	**NA**	**NA**	**NA**	**0.000**	**0.000**	Xie, et al. 2013 [6]
HLA-DPB2 rs2064478	A vs. G	0.22 (0.17–0.30)	4.29 × 10^−29^	GPA	Caucasian	459/1503	NA	NA	**NA**	**NA**	**NA**	**NA**	**0.000**	**0.000**	Xie, et al. 2013 [6]
HLA-DPB2 rs3130215	A vs. G	2.42 (2.08–2.82)	2.37 × 10^−30^	GPA	Caucasian	459/1503	NA	NA	**NA**	**NA**	**NA**	**NA**	**0.000**	**0.000**	Xie, et al. 2013 [6]
HLA-DPB2 rs3117230	C vs. T	0.22 (0.17–0.30)	4.29 × 10^−29^	GPA	Caucasian	459/1503	NA	NA	**NA**	**NA**	**NA**	**NA**	**0.000**	**0.000**	Xie, et al. 2013 [6]
HLA-DPB2 rs1883414	T vs. C	0.53 (0.44–0.64)	1.13 × 10^−11^	GPA	Caucasian	459/1503	0.000	0.009	**0.031**	0.970	**0.000**	**0.005**	**0.000**	**0.039**	Xie, et al. 2013 [6]
HLA-DPB2 rs4713607	A vs. G	0.60 (0.52–0.70)	6.70 × 10^−11^	GPA	Caucasian	459/1503	0.000	0.090	**0.006**	0.849	**0.000**	**0.001**	**0.000**	**0.027**	Xie, et al. 2013 [6]
HLA-DPB2 rs3129274	G vs. A	1.56 (1.34–1.82)	1.35 × 10^−8^	GPA	Caucasian	459/1503	0.000	0.309	**0.036**	0.974	**0.000**	**0.048**	**0.002**	**0.708**	Xie, et al. 2013 [6]
HLA-DPB2 rs3117016	T vs. C	0.48 (0.41–0.57)	1.09 × 10^−17^	GPA	Caucasian	459/1503	NA	NA	**NA**	**NA**	**NA**	**NA**	**0.000**	**0.000**	Xie, et al. 2013 [6]
HLA-DPB2 rs3117008	T vs. C	0.60 (0.51-0.70)	4.90 × 10^−11^	GPA	Caucasian	459/1503	0.000	0.090	**0.006**	0.849	**0.000**	**0.001**	**0.000**	**0.027**	Xie, et al. 2013 [6]
HLA-DPB2 rs3117004	C vs. T	0.57 (0.48–0.68)	1.90 × 10^−10^	GPA	Caucasian	459/1503	0.000	0.041	**0.034**	0.972	**0.000**	**0.010**	**0.000**	**0.161**	Xie, et al. 2013 [6]
HLA-DPB2 rs6901221	C vs. A	0.42 (0.32–0.55)	6.08 × 10^−11^	GPA	Caucasian	459/1503	0.000	0.000	0.475	0.999	**0.001**	0.423	**0.002**	**0.651**	Xie, et al. 2013 [6]
COL11A2 rs986521	C vs. T	1.85 (1.57–2.16)	2.91 × 10^−14^	GPA	Caucasian	459/1503	0.000	0.004	**0.000**	0.246	**0.000**	**0.000**	**0.000**	**0.000**	Xie, et al. 2013 [6]
COL11A2 rs2855430	T vs. C	0.33 (0.24–0.45)	3.28 × 10^−13^	GPA	Caucasian	459/1503	0.000	0.000	0.505	0.999	**0.001**	0.357	**0.000**	**0.237**	Xie, et al. 2013 [6]
COL11A2 rs2855425	C vs. T	1.80 (1.54–2.11)	7.77 × 10^−14^	GPA	Caucasian	459/1503	0.000	0.012	**0.001**	0.594	**0.000**	**0.000**	**0.000**	**0.000**	Xie, et al. 2013 [6]
COL11A2 rs2855459	T vs. C	0.32 (0.23–0.44)	2.14 × 10^−13^	GPA	Caucasian	459/1503	0.000	0.000	0.548	0.999	**0.001**	0.427	**0.000**	**0.287**	Xie, et al. 2013 [6]
RXRB rs6531	C vs. T	1.80 (1.54–2.11)	8.48 × 10^−14^	GPA	Caucasian	459/1503	0.000	0.012	**0.001**	0.594	**0.000**	**0.000**	**0.000**	**0.000**	Xie, et al. 2013 [6]
HSD17B8 rs439205	T vs. C	0.31 (0.24–0.39)	3.51 × 10^−23^	GPA	Caucasian	459/1503	NA	NA	**NA**	**NA**	**NA**	**NA**	**0.000**	**0.000**	Xie, et al. 2013 [6]
HSD17B8 rs421446	C vs. T	0.39 (0.31–0.48)	8.90 × 10^−20^	GPA	Caucasian	459/1503	NA	NA	**NA**	**NA**	**NA**	**NA**	**0.000**	**0.000**	Xie, et al. 2013 [6]
RING1 rs213213	A vs. G	1.83 (1.57–2.14)	6.98 × 10^−15^	GPA	Caucasian	459/1503	0.000	0.006	**0.001**	0.375	**0.000**	**0.000**	**0.000**	**0.000**	Xie, et al. 2013 [6]
RING1 rs213212	G vs. T	1.85 (1.58–2.17)	7.63 × 10^−15^	GPA	Caucasian	459/1503	0.000	0.005	**0.001**	0.439	**0.000**	**0.000**	**0.000**	**0.000**	Xie, et al. 2013 [6]
COBL rs1949829	T vs. C	2.19 (1.68–2.86)	3.58 × 10^−9^	GPA	Caucasian	459/1503	0.000	0.003	0.632	0.999	**0.003**	0.759	**0.017**	0.946	Xie, et al. 2013 [6]
CCDC86 rs595018	A vs. G	1.61 (1.36–1.90)	2.74 × 10^−8^	GPA	Caucasian	459/1503	0.000	0.201	**0.064**	0.986	**0.000**	**0.080**	**0.003**	**0.764**	Xie, et al. 2013 [6]
WSCD1 rs7503953	A vs. C	1.72 (1.44–2.06)	1.39 × 10^−9^	GPA	Caucasian	459/1503	0.000	0.068	**0.076**	0.988	**0.000**	**0.053**	**0.001**	**0.555**	Xie, et al. 2013 [6]
**SNPs Reported Non-Significant (5.00 × 10^−8^ ≤ *p* < 1.00 × 10^−4^)**															
TCEB3 rs2076346	C vs. T	1.46 (1.24–1.73)	8.62 × 10^−6^	GPA	Caucasian	459/1503	0.012	0.623	0.512	0.999	**0.019**	0.952	**0.433**	0.999	Xie, et al. 2013 [6]
DAB1 rs264036	C vs. T	0.72 (0.62–0.85)	9.82 × 10^−5^	GPA	Caucasian	459/1503	0.042	0.818	0.713	1.000	**0.113**	0.992	0.807	1.000	Xie, et al. 2013 [6]
DAB1 rs542873	T vs. C	1.37 (1.18–1.59)	4.70 × 10^−5^	GPA	Caucasian	459/1503	0.041	0.884	0.457	0.999	**0.037**	0.975	**0.601**	0.999	Xie, et al. 2013 [6]
DAB1 rs197644	G vs. A	1.37 (1.18–1.59)	3.89 × 10^−5^	GPA	Caucasian	459/1503	0.041	0.884	0.457	0.999	**0.037**	0.975	**0.601**	0.999	Xie, et al. 2013 [6]
LPHN2 rs11579502	C vs. T	1.56 (1.28–1.90)	9.43 × 10^−6^	GPA	Caucasian	459/1503	0.005	0.348	0.684	1.000	**0.027**	0.966	**0.458**	0.999	Xie, et al. 2013 [6]
NCKAP5 rs1134119	C vs. T	1.79 (1.38–2.32)	8.96 × 10^−6^	GPA	Caucasian	459/1503	0.001	0.091	0.896	1.000	**0.106**	0.992	**0.658**	0.999	Xie, et al. 2013 [6]
NCKAP5 rs7585252	G vs. A	1.35 (1.16–1.57)	9.69 × 10^−5^	GPA	Caucasian	459/1503	0.063	0.914	0.607	0.999	**0.096**	0.991	**0.785**	1.000	Xie, et al. 2013 [6]
NEK10 rs1579900	T vs. G	1.51 (1.24–1.85)	3.97 × 10^−5^	GPA	Caucasian	459/1503	0.013	0.474	0.840	1.000	**0.128**	0.993	0.800	1.000	Xie, et al. 2013 [6]
CTNNB1 rs9842536	T vs. C	1.46 (1.23–1.74)	1.63 × 10^−5^	GPA	Caucasian	459/1503	0.014	0.619	0.624	0.999	**0.037**	0.974	**0.575**	0.999	Xie, et al. 2013 [6]
C3orf58 rs1512779	C vs. A	0.72 (0.61–0.84)	2.86 × 10^−5^	GPA	Caucasian	459/1503	0.032	0.836	0.484	0.999	**0.034**	0.972	**0.579**	0.999	Xie, et al. 2013 [6]
PLSCR4 rs7628805	A vs. C	1.41 (1.18–1.67)	9.72 × 10^−5^	GPA	Caucasian	459/1503	0.031	0.763	0.691	1.000	**0.083**	0.989	**0.754**	1.000	Xie, et al. 2013 [6]
ST6GAL1 rs10513807	G vs. A	0.71 (0.61–0.83)	9.49 × 10^−6^	GPA	Caucasian	459/1503	0.022	0.785	0.436	0.999	**0.021**	0.956	**0.471**	0.999	Xie, et al. 2013 [6]
KIAA0746 rs4269167	T vs. C	0.73 (0.63–0.86)	7.72 × 10^−5^	GPA	Caucasian	459/1503	0.057	0.861	0.747	1.000	**0.163**	0.995	0.858	1.000	Xie, et al. 2013 [6]
DCTD rs4862110	C vs. T	1.63 (1.36–1.94)	5.00 × 10^−8^	GPA	Caucasian	459/1503	0.000	0.175	**0.118**	0.993	**0.000**	**0.178**	**0.007**	0.878	Xie, et al. 2013 [6]
OSMR rs357291	C vs. A	0.72 (0.62–0.84)	2.81 × 10^−5^	GPA	Caucasian	459/1503	0.032	0.836	0.484	0.999	**0.034**	0.972	**0.579**	0.999	Xie, et al. 2013 [6]
SEMA6A rs26595	C vs. T	0.74 (0.63–0.86)	9.58 × 10^−5^	GPA	Caucasian	459/1503	0.061	0.913	0.586	0.999	**0.086**	0.989	**0.766**	1.000	Xie, et al. 2013 [6]
GRIA1 rs10515687	T vs. C	1.56 (1.25–1.94)	8.11 × 10^−5^	GPA	Caucasian	459/1503	0.009	0.362	0.874	1.000	**0.150**	0.994	0.811	1.000	Xie, et al. 2013 [6]
WWC1 rs3853242	G vs. A	0.74 (0.63–0.86)	8.55 × 10^−5^	GPA	Caucasian	459/1503	0.061	0.913	0.586	0.999	**0.086**	0.989	**0.766**	1.000	Xie, et al. 2013 [6]
ERGIC1 rs1564259	A vs. G	0.69 (0.58–0.83)	7.19 × 10^−5^	GPA	Caucasian	459/1503	0.023	0.642	0.785	1.000	**0.114**	0.992	**0.797**	1.000	Xie, et al. 2013 [6]
ERGIC1 rs1006721	C vs. T	0.69 (0.58–0.83)	6.22 × 10^−5^	GPA	Caucasian	459/1503	0.023	0.642	0.785	1.000	**0.114**	0.992	**0.797**	1.000	Xie, et al. 2013 [6]
OFCC1 rs9358619	A vs. G	1.45 (1.21–1.74)	4.90 × 10^−5^	GPA	Caucasian	459/1503	0.021	0.642	0.756	1.000	**0.092**	0.990	**0.762**	1.000	Xie, et al. 2013 [6]
HLA-DMA rs3135029	A vs. C	1.61 (1.28–2.02)	4.55 × 10^−5^	GPA	Caucasian	459/1503	0.006	0.270	0.875	1.000	**0.125**	0.993	**0.764**	1.000	Xie, et al. 2013 [6]
HLA-DOA rs176248	T vs. C	0.70 (0.58–0.83)	8.16 × 10^−5^	GPA	Caucasian	459/1503	0.022	0.713	0.644	0.999	**0.054**	0.983	**0.668**	1.000	Xie, et al. 2013 [6]
HLA-DOA rs206762	C vs. T	1.36 (1.17–1.58)	5.92 × 10^−5^	GPA	Caucasian	459/1503	0.051	0.900	0.534	0.999	**0.061**	0.985	**0.703**	1.000	Xie, et al. 2013 [6]
HLA-DOA rs9296068	G vs. T	0.68 (0.58–0.81)	7.49 × 10^−6^	GPA	Caucasian	459/1503	0.011	0.588	0.578	0.999	**0.026**	0.964	**0.491**	0.999	Xie, et al. 2013 [6]
HLA-DPB2 rs1810472	G vs. A	0.65 (0.54–0.77)	1.06 × 10^−6^	GPA	Caucasian	459/1503	0.002	0.385	0.235	0.997	**0.002**	0.618	**0.062**	0.985	Xie, et al. 2013 [6]
HLA-DPB2 rs3117035	A vs. G	0.72 (0.61–0.84)	2.50 × 10^−5^	GPA	Caucasian	459/1503	0.032	0.836	0.484	0.999	**0.034**	0.972	**0.579**	0.999	Xie, et al. 2013 [6]
COL11A2 rs2235498	T vs. C	0.67 (0.55–0.81)	3.86 × 10^−5^	GPA	Caucasian	459/1503	0.012	0.521	0.744	1.000	**0.063**	0.985	**0.678**	1.000	Xie, et al. 2013 [6]
WDR46 rs3130257	T vs. C	1.63 (1.32–2.01)	5.61 × 10^−6^	GPA	Caucasian	459/1503	0.002	0.218	0.700	1.000	**0.022**	0.957	**0.365**	0.998	Xie, et al. 2013 [6]
DAXX rs211474	T vs. C	0.71 (0.60–0.84)	6.37 × 10^−5^	GPA	Caucasian	459/1503	0.031	0.769	0.679	1.000	**0.078**	0.988	**0.744**	1.000	Xie, et al. 2013 [6]
KIFC1 rs211452	C vs. T	0.65 (0.55–0.76)	1.98 × 10^−7^	GPA	Caucasian	459/1503	0.001	0.375	**0.067**	0.986	**0.000**	**0.150**	**0.008**	0.895	Xie, et al. 2013 [6]
SYNGAP1 rs211456	A vs. C	0.72 (0.61–0.84)	3.57 × 10^−5^	GPA	Caucasian	459/1503	0.032	0.836	0.484	0.999	**0.034**	0.972	**0.579**	0.999	Xie, et al. 2013 [6]
SYNGAP1 rs2247385	G vs. A	0.69 (0.59–0.81)	5.14 × 10^−6^	GPA	Caucasian	459/1503	0.011	0.663	0.353	0.998	**0.009**	0.896	**0.271**	0.997	Xie, et al. 2013 [6]
FLJ43752 rs210120	G vs. A	0.72 (0.62–0.84)	2.58 × 10^−5^	GPA	Caucasian	459/1503	0.032	0.836	0.484	0.999	**0.034**	0.972	**0.579**	0.999	Xie, et al. 2013 [6]
BCKDHB rs515347	G vs. A	1.70 (1.32–2.18)	3.13 × 10^−5^	GPA	Caucasian	459/1503	0.003	0.162	0.905	1.000	**0.151**	0.994	**0.769**	1.000	Xie, et al. 2013 [6]
TCBA1 rs6924068	G vs. A	1.40 (1.20–1.63)	1.37 × 10^−5^	GPA	Caucasian	459/1503	0.023	0.813	0.382	0.998	**0.018**	0.947	**0.429**	0.999	Xie, et al. 2013 [6]
MAGI2 rs3779312	A vs. G	1.50 (1.26–1.78)	5.26 × 10^−6^	GPA	Caucasian	459/1503	0.005	0.500	0.392	0.998	**0.007**	0.873	**0.214**	0.996	Xie, et al. 2013 [6]
CUTL1 rs1734729	T vs. C	1.41 (1.20–1.67)	4.54 × 10^−5^	GPA	Caucasian	459/1503	0.031	0.763	0.691	1.000	**0.083**	0.989	**0.754**	1.000	Xie, et al. 2013 [6]
DPP6 rs4726422	G vs. A	0.74 (0.63–0.86)	7.74 × 10^−5^	GPA	Caucasian	459/1503	0.061	0.913	0.586	0.999	**0.086**	0.989	**0.766**	1.000	Xie, et al. 2013 [6]
LOC441376 rs3019885	G vs. T	1.44 (1.24–1.67)	1.90 × 10^−6^	GPA	Caucasian	459/1503	0.008	0.705	**0.151**	0.994	**0.002**	0.667	**0.092**	0.990	Xie, et al. 2013 [6]
SLC30A8 rs1793729	C vs. T	0.68 (0.57–0.80)	5.68 × 10^−6^	GPA	Caucasian	459/1503	0.007	0.594	0.317	0.998	**0.006**	0.847	**0.194**	0.996	Xie, et al. 2013 [6]
SLC30A8 rs1695715	T vs. C	0.69 (0.58–0.82)	1.57 × 10^−5^	GPA	Caucasian	459/1503	0.016	0.652	0.610	0.999	**0.037**	0.975	**0.582**	0.999	Xie, et al. 2013 [6]
KCNK9 rs2447406	T vs. C	1.58 (1.27–1.97)	3.25 × 10^−5^	GPA	Caucasian	459/1503	0.007	0.322	0.869	1.000	**0.130**	0.993	**0.781**	1.000	Xie, et al. 2013 [6]
C9orf66 rs584922	T vs. C	0.70 (0.60–0.82)	1.30 × 10^−5^	GPA	Caucasian	459/1503	0.015	0.727	0.392	0.998	**0.013**	0.932	**0.365**	0.998	Xie, et al. 2013 [6]
C9orf93 rs1341740	T vs. C	1.52 (1.23–1.87)	9.73 × 10^−5^	GPA	Caucasian	459/1503	0.013	0.450	0.855	1.000	**0.142**	0.994	0.814	1.000	Xie, et al. 2013 [6]
LRRN6C rs10491888	G vs. A	1.54 (1.25–1.91)	5.75 × 10^−5^	GPA	Caucasian	459/1503	0.012	0.405	0.880	1.000	**0.173**	0.995	0.837	1.000	Xie, et al. 2013 [6]
PAEP rs705669	G vs. A	0.69 (0.57–0.83)	8.53 × 10^−5^	GPA	Caucasian	459/1503	0.023	0.642	0.785	1.000	**0.114**	0.992	**0.797**	1.000	Xie, et al. 2013 [6]
NEUROG3 rs731573	T vs. C	1.44 (1.20–1.72)	9.40 × 10^−5^	GPA	Caucasian	459/1503	0.022	0.674	0.722	1.000	**0.079**	0.988	**0.738**	1.000	Xie, et al. 2013 [6]
TMPO rs2216021	C vs. T	1.36 (1.17–1.57)	6.14 × 10^−5^	GPA	Caucasian	459/1503	0.044	0.909	0.382	0.998	**0.029**	0.967	**0.544**	0.999	Xie, et al. 2013 [6]
TMPO rs2011247	C vs. T	0.73 (0.63–0.85)	4.84 × 10^−5^	GPA	Caucasian	459/1503	0.044	0.879	0.534	0.999	**0.054**	0.983	**0.680**	1.000	Xie, et al. 2013 [6]
FGF9 rs2031421	T vs. G	1.44 (1.21–1.72)	4.01 × 10^−5^	GPA	Caucasian	459/1503	0.022	0.674	0.722	1.000	**0.079**	0.988	**0.738**	1.000	Xie, et al. 2013 [6]
DOK5 rs6023640	T vs. G	1.40 (1.18–1.66)	8.98 × 10^−5^	GPA	Caucasian	459/1503	0.038	0.786	0.740	1.000	**0.121**	0.993	0.815	1.000	Xie, et al. 2013 [6]
PDE9A rs2269127	A vs. G	1.65 (1.36–2.01)	4.06 × 10^−7^	GPA	Caucasian	459/1503	0.001	0.172	0.457	0.999	**0.004**	0.793	**0.094**	0.990	Xie, et al. 2013 [6]

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
