# Peer review of "Genetic Variants in Antineutrophil Cytoplasmic Antibody-Associated Vasculitis: A Bayesian Approach and Systematic Review"

_jcm, 2019, doi:10.3390/jcm8020266_

Reviewer 1 Report

This manuscript reviews 12 published articles out of 241 potential articles (5% review rate). In some sense, this is a meta-analysis of just 12 articles. I do have an initial sense that 12 articles may be too slim - to this, the authors should categorize why the other 95% are not used. I do not find it convincing that 95% of the potential articles are left out from a systematic review - is it possible that the criteria are too strict.

I do have to urge the authors to clean up the draft - accept all changes - before submission as it is too unprofessional to submit with all the changes.

Author Response

1)This manuscript reviews 12 published articles out of 241 potential articles (5% review rate). In some sense, this is a meta-analysis of just 12 articles. I do have an initial sense that 12 articles may be too slim - to this, the authors should categorize why the other 95% are not used. I do not find it convincing that 95% of the potential articles are left out from a systematic review - is it possible that the criteria are too strict. 

We are very thankful for your interest and the comments raised to improve the manuscript, which have been taken into account during revision of our manuscript.

ANCA-associated vasculitis compromises a group of rare disease. Large genome wide association studies and observational studies were lacking until recently. The aim was to summarize the relevant literature, and with a combined analysis identifying novel pathways which might pave the way for more research in the field. There are several genetic studies on the way which might corroborate our findings. In line with this publication, we have worked on similar projects already published in the past (Jeong DY, et al. Autoimmun Rev 2018; 17:553-566; Park JH, et al. Mol Neurobiol 2018; 55:5672-5688).

 We described the search strategy as follows: At the end of the systematic search, 241 articles were identified, 13 were screened in detail, and 7 were selected for analysis [4-6,8-11]. (Figure 1, Supplementary table A.2). The inclusion criteria were (1) genetic meta-analysis or GWAS providing information of odds ratios (OR), 95% confidence intervals (CI) in AAV, and (2) studies written in English and (3) most recent larger meta-analyses if those are overlapped for same genetic variant. The exclusion criteria were (1) studies not related to genetic polymorphisms or AAV, and (2) articles not providing accurate data sets (i.e. review articles) and older meta-analyses if those are overlapped for same genetic variant. 

To explain more precisely, our inclusion criteria were to re-analyze the meta-analysis and GWAS articles on genetic polymorphisms of AAV. 241 articles were not potential articles for inclusion, because they are found to be other diseases (not AAV), or they were on AAV, but were not genetic polymorphism articles, or review articles which do not contain the precise genetic data. When there are overlapped meta-analysis on same polymorphism sites, we only selected only most-updated last meta-analysis which contains largest number of patients. In addition, if GWAS data are used in meta-analysis as individual studies, we selected only the final meta-analysis results for the candidate polymorphisms. We also excluded 3 meta-analyses which showed no association of polymorphisms, because the aim of our study was to investigate whether genetic variants which are claimed as significant (p<0.05) in="" previous="" meta-analysis="" or="" gwas="" are="" really="" significant="" true="" by="" applying="" a="" bayesian="" approach="" results="" with="" p-value="">0.05 are always found to be not noteworthy by Bayesian approach).

 2)I do have to urge the authors to clean up the draft - accept all changes - before submission as it is too unprofessional to submit with all the changes.

Manuscript ID jcm-448361 was resubmitted from jcm-400485 and the journal proposes that revisions are made to the original submission. “Any revisions should be clearly highlighted, for example using the Track Changes function in Microsoft Word, so that they are easily visible to the editors and reviewers”. We hope that this is a sufficient answer for the “non cleaned-up” draft of the resubmission.

Reviewer 2 Report

The paper by Lee et al provides a new view on the genetic analyses that have taken place in ANCA vasculitis over the last years by means of re-analysis. The use of a Bayesian approach is novel and resulted in interesting findings that when analyzed by GO analysis, support certain thoughts on pathophysiology such as the INFg pathway. Unofrtunately the data are partially limited due to the absence of 95% confidence intervals of earlier analyses. This has been stated in the discussion. Overall I think it is an interesting paper for the vasculitis community with data resulting from a sound statistical re-analysis. 

Major comments:

- discussion: is it possible to construe a single pathophysiological mechanism using all noteworthy genes or is ANCA vasculitis a multigenic disease group? 

- discussion: in your opinion, does the IFNg pathway only play a significant role in PR3-ANCA vasculitis or a similar role in MPO-ANCA vasculitis? 

- tables 1, 2 and 3: for publication I would suggest including only the noteworthy genetic variants in order to make the table actually readable, the others can be included in the supplementary files.

- In the introduction (line 70) the authors mention the differences between PR3-ANCA and MPO-ANCA and/or GPA and MPA. I want to advocate a citation of our Maastricht group review on that matter in J Am Soc Nephrol (2015; 26:2314; https://doi.org/10.1681/ASN.2014090903). 

Minor comments: 

- Experimental section: construction of PPI (line 143): in the STRING database, PPIs associated with AAV-mapped genes included AAV, GPA,PR3-ANCA and MPO-ANCA. I was wondering: why was MPA not included? 

- introduction line 65: "granulomatous" should be "granulomatosis".

Author Response

The paper by Lee et al provides a new view on the genetic analyses that have taken place in ANCA vasculitis over the last years by means of re-analysis. The use of a Bayesian approach is novel and resulted in interesting findings that when analyzed by GO analysis, support certain thoughts on pathophysiology such as the INFg pathway. Unofrtunately the data are partially limited due to the absence of 95% confidence intervals of earlier analyses. This has been stated in the discussion. Overall I think it is an interesting paper for the vasculitis community with data resulting from a sound statistical re-analysis. 

We are very thankful for your interest and the comments raised to improve the manuscript, which have been taken into account during revision of our manuscript.

Major comments:

1- discussion: is it possible to construe a single pathophysiological mechanism using all noteworthy genes or is ANCA vasculitis a multigenic disease group? 

From our findings we need to conclude that AAV is indeed a multigenic disease, since we know that not only genes are implicated but also the environment (seasonal differences) and a two-hit hypothesis was recently again discussed (Cohen Tervaert, Editorial series, Nephrol Dial Transplant 2019, in press). We have made a respective statement highlighting this in the Discussion.

2- discussion: in your opinion, does the IFNg pathway only play a significant role in PR3-ANCA vasculitis or a similar role in MPO-ANCA vasculitis? 

Supplementary Figure A.3. highlights analysis of different GO enrichment analysis and reveals an involvement (comparable) of the IFN-g pathway in both, PR3-ANCA and MPO-ANCA vasculitis. This has been stated in the Discussion.

3- tables 1, 2 and 3: for publication I would suggest including only the noteworthy genetic variants in order to make the table actually readable, the others can be included in the supplementary files.

According to the comments, we moved table 1-4 to supplemental tables and presented only the noteworthy genetic variants in tables.

4- In the introduction (line 70) the authors mention the differences between PR3-ANCA and MPO-ANCA and/or GPA and MPA. I want to advocate a citation of our Maastricht group review on that matter in J Am Soc Nephrol (2015; 26:2314; https://doi.org/10.1681/ASN.2014090903). 

We are well aware of this excellent review. Since there is mounting evidence to split ANCA-associated vasculitis in PR3-ANCA and MPO-ANCA vasculitis, our work seems to have additional power. We have added the proposed reference according to the comment.

Minor comments: 

5- Experimental section: construction of PPI (line 143): in the STRING database, PPIs associated with AAV-mapped genes included AAV, GPA,PR3-ANCA and MPO-ANCA. I was wondering: why was MPA not included? 

If you see the table 1, we have not identified significant genetic associations with microscopic polyangiitis (MPA). Thus, this was not incorporated in our analysis.

6- introduction line 65: "granulomatous" should be "granulomatosis".

Many thanks for bringing this typographical error to our attention and we corrected it

Reviewer 3 Report

The authors presented a great review about genetic variants in antineutrophil cytoplasmic antibody associated vasculitis. However it is very well written manuscript, however, I have some minor correcyions regarding language which should be fixed before any further progress.

Here are the minor concerns:

67 The clinical manifestations of the disease types can vary widely; most patients have signs of lung…. Add an article (a/ the) before lung.

68 previous observational studies employing single nucleotide polymorphisms (SNPs) revealed number.. Add an article (a/ the) before number.

 86 the statistical analyses leads to an increased likelihood of finding false-positive associations. Singular verb leads doesn’t go with plural analyses. Consider changing the verb.

 134 This study presents both FPRP and BFDP methods because genetic epidemiologists and…missing preposition after because

 137 and BFDP, we provide readers options for interpreting noteworthiness. Missing preposition after readers.

 194 the PR3-ANCA and MPO-ANCA vasculitis associated SNPs showedrepresented their closer.. space after showed.

 209 (Figure 4). Other subtypes were also identified with MHC class II related GO terms and IFN-γ.. Missing hyphen between II and related.

Author Response

The authors presented a great review about genetic variants in antineutrophil cytoplasmic antibody associated vasculitis. However, it is very well written manuscript, however, I have some minor corrections regarding language which should be fixed before any further progress.

Response: We are very grateful that you like our manuscript and for your comments to correct for typographical errors / grammatic errors.

Here are the minor concerns:

67 The clinical manifestations of the disease types can vary widely; most patients have signs of lung…. Add an article (a/ the) before lung.R: This has been changed according to your comment.

 78 previous observational studies employing single nucleotide polymorphisms (SNPs) revealed number. Add an article (a/ the) before number.

 R: This has already been changed, thanking for bringing it to our attention.

 86 the statistical analyses leads to an increased likelihood of finding false-positive associations. Singular verb leads doesn’t go with plural analyses. Consider changing the verb.

 R: Again, this is important and has been changed accordingly.

 134 This study presents both FPRP and BFDP methods because genetic epidemiologists and…missing preposition after because

 R: Thank you very much for this comment.

 137 and BFDP, we provide readers options for interpreting noteworthiness. Missing preposition after readers.

 R: This has been changed accordingly.

 194 the PR3-ANCA and MPO-ANCA vasculitis associated SNPs showedrepresented their closer.. space after showed.

 R: This has been changed. Many thanks.

 209 (Figure 4). Other subtypes were also identified with MHC class II related GO terms and IFN-γ.. Missing hyphen between II and related.

 This has already been deleted.